# Semilicoisoflavone B Induces Apoptosis of Oral Cancer Cells by Inducing ROS Production and Downregulating MAPK and Ras/Raf/MEK Signaling

**DOI:** 10.3390/ijms24054505

**Published:** 2023-02-24

**Authors:** Ming-Ju Hsieh, Hsin-Yu Ho, Yu-Sheng Lo, Chia-Chieh Lin, Yi-Ching Chuang, Mosleh Mohammad Abomughaid, Ming-Chang Hsieh, Mu-Kuan Chen

**Affiliations:** 1Oral Cancer Research Center, Changhua Christian Hospital, Changhua 500, Taiwan; 2Ph.D. Program in Tissue Engineering and Regenerative Medicine, College of Medicine, National Chung Hsing University, Taichung 402, Taiwan; 3Graduate Institute of Biomedical Sciences, China Medical University, Taichung 404, Taiwan; 4Department of Medical Laboratory Sciences, College of Applied Medical Sciences, University of Bisha, Bisha 61922, Saudi Arabia; 5School of Medical Laboratory and Biotechnology, Chung Shan Medical University, Taichung 40201, Taiwan; 6Department of Clinical Laboratory, Chung Shan Medical University Hospital, Taichung 40201, Taiwan; 7Department of Post-Baccalaureate Medicine, College of Medicine, National Chung Hsing University, Taichung 402, Taiwan; 8Department of Otorhinolaryngology, Head and Neck Surgery, Changhua Christian Hospital, Changhua 500, Taiwan

**Keywords:** oral squamous cell carcinoma, semilicoisoflavone B, MAPK, Ras/Raf/MEK, ROS, survivin

## Abstract

Oral squamous cell carcinoma (OSCC) is the sixth most common type of cancer worldwide. Despite advancement in treatment, advanced-stage OSCC is associated with poor prognosis and high mortality. The present study aimed to investigate the anticancer activities of semilicoisoflavone B (SFB), which is a natural phenolic compound isolated from *Glycyrrhiza species*. The results revealed that SFB reduces OSCC cell viability by targeting cell cycle and apoptosis. The compound caused cell cycle arrest at the G2/M phase and downregulated the expressions of cell cycle regulators including cyclin A and cyclin-dependent kinase (CDK) 2, 6, and 4. Moreover, SFB induced apoptosis by activating poly-ADP-ribose polymerase (PARP) and caspases 3, 8, and 9. It increased the expressions of pro-apoptotic proteins Bax and Bak, reduced the expressions of anti-apoptotic proteins Bcl-2 and Bcl-xL, and increased the expressions of the death receptor pathway protein Fas cell surface death receptor (FAS), Fas-associated death domain protein (FADD), and TNFR1-associated death domain protein (TRADD). SFB was found to mediate oral cancer cell apoptosis by increasing reactive oxygen species (ROS) production. The treatment of the cells with N-acetyl cysteine (NAC) caused a reduction in pro-apoptotic potential of SFB. Regarding upstream signaling, SFB reduced the phosphorylation of AKT, ERK1/2, p38, and JNK1/2 and suppressed the activation of Ras, Raf, and MEK. The human apoptosis array conducted in the study identified that SFB downregulated survivin expression to induce oral cancer cell apoptosis. Taken together, the study identifies SFB as a potent anticancer agent that might be used clinically to manage human OSCC.

## 1. Introduction

Oral squamous cell carcinoma (OSCC), commonly known as oral cancer, is the sixth most common cancer and a leading public health concern worldwide. About 263,000 new cases of oral cancer and 127,000 deaths are reported globally in each year. It is the third most commonly detected cancer in India, which accounts for 40% of cancer-related deaths [1]. Tobacco use and alcohol consumption are two leading causes of OSCC, which accounts for about 95% of all head and neck cancers [2]. Although the survival rate of OSCC is about 50%, a successful management of the condition is possible with early detection and treatment. 

Surgery with or without radiotherapy and chemotherapy is conventionally used to treat oral cancer patients [3]. However, these therapeutic approaches come with many detrimental side-effects. Moreover, despite many advances, these approaches have failed to increase the survival rate beyond 50% [4]. The disease-free and overall survival of patients with OSCC depends on many factors, including the site of lesion, stage of detection, tumor size, nodal status, marginal status, and treatment modality [5,6]. Tumor relapse is another major concern, which occurs in 25-30% of early-stage oral cancer patients and 50-60% of advance-stage patients [7]. These tumors frequently develop drug-resistant cells, and thus are associated with poor prognosis [8]. Cancer immunotherapy with immune checkpoint inhibitors has shown promising outcome in recurrent and metastatic tumors [3]. High tumor mutational burden is considered to be an effective predictive marker for assessing the success rate of immunotherapy because of the presence of higher numbers of immunogenic neoantigens. However, because of tumor heterogeneity, a universal threshold of tumor mutational burden might not serve as an accurate marker to predict the efficacy of immunotherapy at the individual-level [9]. Thus, identification and validation of suitable predictive markers is highly needed in order to obtain the optimal therapeutic benefits.

To overcome the shortcomings of conventional cancer therapies and targeted immunotherapy, researchers are putting considerable effort to identify potential phytochemicals for cancer management [10,11,12]. Phytochemicals are natural plant-derived compounds with many beneficial health effects, including anti-inflammatory, antioxidant, immunomodulatory, and anticancer properties [13]. In cancer treatment, phytochemicals are known to primarily work by regulating various signal transduction pathways related to cancer cell proliferation and metastasis [10]. Many phytochemicals with anticancer properties, including curcumin, berberine, epigallocatechin gallate, resveratrol, quercetin, and sulforaphane, are currently under investigation in clinical trials [12].

*Glycyrrhiza* (licorice) is a widely used traditional medicinal plant with many health benefits [14,15]. About 30 species have been identified worldwide, with *Glycyrrhiza glabra* and *Glycyrrhiza uralensis* being the most commonly distributed species [14]. More than 400 bioactive compounds including 300 flavonoids have been isolated from *Glycyrrhiza* species. Flavonoids are the major compounds with anti-inflammatory, antioxidant, antiviral, and anticancer activities [16]. Regarding anticancer activity, *Glycyrrhiza glabra*-derived flavonoids have been found to inhibit cancer cell growth by inducing cell cycle arrest and apoptosis [17]. The compounds have been found to target a variety of signaling pathways to exert anticancer activities against many cancer types, including gastric cancer [18], breast cancer [19], melanoma [20], and OSCC [21]. 

Semilicoisoflavone B (SFB) is a natural isoprenoid-substituted phenolic compound isolated from *Glycyrrhiza glabra* and *Glycyrrhiza uralensis* [22]. The compound has been found to act as a peroxisome proliferator activated receptor (PPAR) agonist to reduce secretion of amyloid beta plaques, which is the major hallmark of Alzheimer’s disease [23]. Moreover, the compound has been found to prevent osmotic stress in hyperglycemia [24,25]. The tyrosinase inhibitory activity of the compound has also been reported in the literature [25]. To the best of our knowledge, no studies have so far investigated the anticancer properties of SFB. In the present, we have investigated the anticancer effects and mode of action of SFB in human OSCC.

## 2. Results

### 2.1. Cytotoxic Effect of SFB in Oral Cancer Cells

We firstly investigated whether SFB treatment reduces viability of human oral cancer cells. Semilicoisoflavone B is a member of the class of 7-hydroxyisoflavones, which 2′,2′-dimethyl-2′H,4H-3,6′-bichromen-4-one substituted by hydroxy groups at positions 5, 7 and 8′ (Figure 1A). We treated four different OSCC cell lines with three different doses (25 µM, 50 µM, and 100 µM) of SFB at three timepoints (24 h, 48 h, and 72 h) and performed MTT assay, as mentioned in the Materials and Methods section. The cells without any treatment served as experimental controls. As observed in Figure 1B, the SFB treatment significantly and dose-dependently reduced the viability of all tested cell lines. The colony formation assay conducted following the same treatment revealed that all doses of SFB are equally capable of significantly reducing the colony formation ability of oral cancer cells (Figure 1C,D). Overall, these findings indicate that SFB has high potency in preventing oral cancer cell proliferation, which is regarded as one of the basic characteristics of an anticancer compound.

### 2.2. Effect of SFB on Cell Cycle Arrest and Apoptosis in Oral Cancer Cells 

We next investigated whether SFB affects cell viability by inducing cell cycle arrest and apoptosis. We performed cell cycle analysis using two OSCC cell lines that were treated with the same doses of SFB. As observed in Figure 2: A and B, the SFB treatment significantly and dose-dependently reduced the cell cycle rate at the G0/G1 phase and increased it at the S and G2/M phases. The effects were similar in both cell lines. Furthermore, we determined the expression of cell cycle control proteins and observed that the SFB treatment significantly reduced the expressions of cyclin A, CDK2, CDK4, and CDK6 in both cell lines (Figure 2C,D). Taken together, these findings highlight the ability of SFB in inducing cell cycle arrest. Mechanistically, SFB exerted cytotoxic effects by arresting the cell cycle at G2/M phase which resulted in an increased percentage of cells in the G2/M phase together with a reduced distribution in the G0/G1 phase. Next, we investigated whether SFB can affect the apoptotic pathways in OSCC cells. Firstly, we stained the SFB-treated cells with DAPI and assessed nuclear condensation using fluorescence microscopy. As observed in Figure 3A,B, the SFB treatment significantly increased the percentage of nuclear condensation in a dose-dependent manner. Since nuclear condensation and fragmentation is a well-known marker of apoptosis [26], this finding suggests that SFB exerts its cytotoxic effect by inducing apoptosis. Moreover, the analysis of mitochondrial membrane potential revealed that the higher doses of SFB significantly increased the levels of mitochondrial membrane depolarization in both cell lines (Figure 4A,B).

To further verify the pro-apoptotic activity of SFB, we stained SFB-treated cells with Annexin V/PI and performed flow cytometry. The findings revealed that the SFB treatment significantly increased the percentage of apoptotic cells in a dose-dependent manner (Figure 3C,D). Caspases are a family of cellular proteases that play an essential role in controlling apoptosis. Caspases 8 and 9 are considered initiator caspases, and caspases 3, 6, and 7 are considered executioner caspases [27]. We determined the protein expressions of cleaved caspases and cleaved PARP and observed that the SFB treatment significantly increased the expressions of cleaved caspases 3, 8, and 9 and cleaved PARP in both oral cancer cell lines (Figure 2E,F). Next, we determined the expressions of extrinsic and intrinsic apoptotic pathway proteins. As observed in Figure 4C,D, the SFB treatment significantly increased the expressions of pro-apoptotic proteins Bax and Bak and reduced the expressions of anti-apoptotic proteins Bcl-2 and Bcl-xL in both cell lines. These findings indicate the induction of intrinsic pathway following SFB treatment. Regarding extrinsic pathway, the SFB was found to cause a significant induction in Fas, FADD, and TRADD expressions in both cell lines (Figure 4E,F). For the final confirmation, we treated the cells with a pan-caspase inhibitor Z-VAD-FMK in presence or absence of SFB and checked the expressions of cleaved caspase 3 and cleaved PARP. As observed in Figure 4G,H, the combined treatment with SFB and Z-VAD-FMK caused a higher reduction in cleaved caspase 3 and cleaved PARP expressions compared to that caused by the SFB treatment alone. Taken together, these findings indicate that SFB triggers oral cancer cell apoptosis via both intrinsic and extrinsic pathways.

### 2.3. Effect of SFB on Free Radical Production and Apoptosis in Oral Cancer Cells

A low to moderate level of reactive oxygen species (ROS) production in the body is essential for regulating various physiological functions, including cell proliferation, differentiation, migration, and apoptosis. ROS acts a viral signaling molecule in regulation main apoptotic pathways inducted by mitochondria, death receptors, and endoplasmic reticulum [28]. Given the involvement of both intrinsic and extrinsic apoptotic pathways in the study, we thought of investigating whether SFB activates these pathways by triggering ROS production. As observed in Figure 5A,B, the treatment with SFB significantly increased the expression of ROS in both cell lines in a dose- and time-dependent manner. A significantly lower ROS level was observed in cells cotreated with SFB and a ROS inhibitor (N-acetyl cysteine; NAC), compared with that in cells treated with SFB alone (Figure 5C). These findings clearly indicate that SFB induces ROS production in oral cancer cells. When analyzed the apoptotic rate in SFB-treated oral cancer cells in presence and absence of NAC, we observed that the cotreatment with SFB and NAC significantly reduced the percentage of apoptotic cells compared to the SFB treatment alone (Figure 5D). This finding highlights that SFB induces oral cancer cell apoptosis by increasing ROS production and that NAC-induced removal of ROS from the cells mitigates the pro-apoptotic activity of SFB.

### 2.4. Effect of SFB on Cellular Signaling Pathways in Oral Cancer Cells

We next investigated which signaling pathways are directly involved in mediating SFB-induced oral cancer cell apoptosis. We selected three major pathways, including AKT, MAPK, and RAS/MEK pathways, which are directly involved in modulating apoptotic pathways. As observed in Figure 6A,B, the SFB treatment significantly reduced AKT, ERK1/2, p38, and JNK1/2 phosphorylation in both oral cancer cell lines. Furthermore, SFB at higher concentration caused a significant reduction in RAS, RAF, and MEK expression in both cell lines (Figure 6C,D). Taken together, these findings indicate that SFB specifically targeting the AKT, MAPK and RAS/RAF/MEK pathways. 

### 2.5. Effect of SFB on Specific Apoptosis-Related Protein Expression 

We performed human apoptosis array to identify apoptosis-related proteins that are specifically affected by the SFB treatment. As observed in Figure 7A,B, the SFB treatment caused a significant reduction in survivin expression at 100 µM concentration. This effect was further confirmed in two oral cancer cell lines, where SFB was found to reduce survivin expression in a dose-dependent manner (Figure 7C,D). To further evaluate the involvement of survivin, we overexpressed the protein in oral cancer cells and analyzed the expressions of cleaved PARP and cleaved caspase 3 in presence and absence of SFB. As observed in Figure 7E,F, SFB-induced increase in cleaved caspase 3 and cleaved PARP expression was significantly mitigated by survivin overexpression in oral cancer cells. Taken together, these findings indicate that SFB induces cleaved caspase 3 and cleaved PARP expression by suppressing the expression of survivin, which is a member of the inhibitor of apoptosis (IAP) protein family.

## 3. Discussion

In the present study, we investigated the anticancer effects and mode of action of SFB (Figure 1A), which is a phenolic compound derived from medicinal plants belonging to the *Glycyrrhiza* genus. Previous studies have highlighted that SFB acts as a PPAR agonist to ameliorates that progression of Alzheimer’s disease [22]. The compound has also been found to inhibit the growth of the Gram-positive bacterium *Bacillus subtilis* [15,29]. This study is the first of its kind to demonstrate anticancer effect of SFB in OSCC. 

We tested different concentrations of SFB (25 µM, 50µM, and 100 µM) and found that each dose is highly effective in significantly reducing the viability and colony formation ability of OSCC cells (Figure 1B–D). Furthermore, we observed that SFB exerts cytotoxic effects by causing cell cycle arrest at the G2/M phase and inducing apoptosis via both intrinsic and extrinsic pathways (Figure 2, Figure 3 and Figure 4). The compound was found to prevent cell cycle progression by suppressing the expressions of a number of cell cycle regulators, including cyclin A and cyclin-dependent kinases (CDKs) (Figure 2C,D). Regarding apoptosis regulation, the compound was found to activate PARP and caspases 3, 8, and 9 (Figure 2E,F), induce pro-apoptotic proteins Bak and Bax, and suppress anti-apoptotic proteins Bcl-2 and Bcl-xL (Figure 4C,D). Apart from above-mentioned intrinsic apoptotic pathways, the compound induced death receptor-mediated extrinsic apoptotic pathway, as evidenced by increased expressions of Fas, FADD, and TRADD expressions in SFB-treated oral cancer cells (Figure 4E,F).

Our findings are in line with previous studies investigation the anticancer potential of derived bioactive compounds [30,31,32,33]. As with SFB, other prenylated phenolic compounds derived from Glycyrrhiza species have been found induce cancer cell death by inducing cell cycle arrest and caspase-mediated apoptotic pathways [30,34,35]. One recent study has shown that *Glycyrrhiza glabra*-derived bioactive compound glycyrrhizin reduces viability of cervical cancer cells by increasing nuclear condensation and DNA fragmentation. The compound has also been found to increase ROS production, disrupt mitochondrial membrane potential, induce caspase-mediated apoptosis, and causing cell cycle arrest at G0/G1 phase. A significant reduction in glycyrrhizin-mediated cytotoxicity has been observed in cervical cancer cells treated with ROS inhibitor NAC [36]. Similar findings have also been reported by another study investing the anticancer potential of *Glycyrrhiza foetida*-derived bioactive compound amorfrutin C [37]. These findings are in line with the results from our study, which demonstrate that an induction in intracellular ROS level is crucial for SFB-mediated apoptosis of oral cancer cells (Figure 5). In order to see the mode of action of SFB on normal cells, it would be interesting to perform some of the experiments related to cell cycle arrest, apoptosis and ROS on human oral epithelial cells.

Regarding upstream signaling pathways, we observed that SFB suppresses Ras/Raf/MEK and MAPK (ERK1/2, p38, and JNK1/2) signaling pathways to exert its pro-apoptotic effects in oral cancer cells (Figure 6). The suppression of MAPK signaling by *Glycyrrhiza species*-derived bioactive compounds has been documented by previous studies investing anti-inflammatory, anti-proliferative, and anticancer activities of phytochemicals [17,38,39]. One recent study conducting a thorough transcriptome analysis has indicated that the MAPK signaling pathway is one of the crucial targets of Glycyrrhiza-derived compound licochalcone A related to its anticancer activity [40]. The alteration of Ras stability and modulation of Ras-mediated signaling by Glycyrrhiza-derived compounds have also been reported by previous studies [41,42,43]. In agreement with previous study findings, our study provides a detailed overview of signaling pathways affected by SFB in oral cancer cells. 

Importantly, we identified survivin as a major target and critical regulator of SFB-induced apoptosis of oral cancer cells (Figure 7). Survivin is a member of the inhibitor of apoptosis family protein, which prevents cell death by downregulating caspases [44]. In agreement with our findings, some recent studies have shown that Glycyrrhiza-derived compounds licochalcone A and licochalcone C mediate their anticancer activities by downregulating survivin [45,46,47,48,49]. SFB was found to reduce survivin expression in a dose-dependent manner (Figure 7C,D). As observed in Figure 7E,F, SFB-induced increase in cleaved caspase 3 and cleaved PARP expression was significantly mitigated by survivin overexpression in oral cancer cells. These findings indicate that SFB induces oral cancer cell apoptosis by targeting survivin, which is a member of the inhibitory apoptosis (IAP) protein family.

## 4. Materials and Methods

### 4.1. Cell Culture

Human OSCC cell lines SAS, SCC9, OECM-1 and HSC3M3 were selected for experimentation. The cell lines were purchased from the Japanese Collection of Research Bioresource Cell Bank (Shinjuku, Japan). SAS and SCC9 were cultured in Dulbecco’s Modified Eagle Medium (DMEM) (Life Technologies, Grand Island, NY, USA)/F12. OECM-1 was cultured in RPMI 1640. HSC3M3 was cultured in Eagle’s minimal essential medium (MEM). The culture media were supplemented with 10% fetal bovine serum (FBS), 10,000 U/mL penicillin and 10 mg/mL streptomycin. All cell lines were maintained at 37 °C in a humidified atmosphere of 5% CO_2_.

### 4.2. Chemical Treatments

Semilicoisoflavone B (SFB) of ≥98% purity was purchased from ChemFaces (Wuhan, Hubei, China). SFB stock solution (100 mM) was prepared using dimethyl sulfoxide (DMSO, Sigma-Aldrich) and stored at −20°C. The DMSO concentration was less than 0.2% for each experiment. For the treatment with SFB, appropriate amounts of stock solution were administered to the medium to obtain the final experimental doses. Other chemical reagents used in the study including 3-(4,5-dimethylthiazol-2-yl)-2,5-diphenyltetrazolium bromide (MTT), propidium iodide (PI), RNase A, DAPI dye, protease inhibitor cocktail, and phosphatase inhibitor cocktail were purchased from Sigma-Aldrich (St Louis, MO, USA). The primary antibodies were purchased from Cell Signaling Technology (Danvers, MA, USA).

### 4.3. Cell Viability (MTT Assay)

To study the effects of SFB on cell viability, an MTT assay was performed as described previously [50]. Briefly, cells were seeded onto 96-well plates and then treated with different concentrations (0, 25, 50, or 100 μM) of semilicoisoflavone B for 24 h at 37 °C. Medium-diluted MTT was added to each well, and the cells were incubated for 2 h at 37 °C. The formazan crystals were dissolved in DMSO and measured with a microplate reader (BioTek; Winooski, VT, USA) at 570 nm.

### 4.4. Colony Formation Assay

The cells were seeded onto 6-well plates at a density of 5 × 10^3^ cells/well and cultured overnight, followed by incubation with different concentrations of SFB. The incubation medium was changed every 3 days. After two weeks, the colonies were fixed with 4% paraformaldehyde and stained with 0.3% crystal violet solution. The stained colonies were dissolved in DMSO and counted by a stereomicroscope as previously described [51].

### 4.5. Cell Cycle Analysis

Cell cycle analysis was performed as previously described [52]. The cells were seeded onto 6-well plates (2 × 10^5^ cells/well) and treated with various concentrations of SFB for 24 h. The cells were harvested by centrifugation and fixed in 70% ethanol at −20 °C for 12 h. The fixed cells were stained with PI buffer (4 mg/mL PI, 1% Triton X-100, 0.5 mg/mL RNase A in PBS) for 30 min in the dark at room temperature. The cell cycle distribution was analyzed by BD Accuri C6 Plus flow cytometry (BD Biosciences, San Jose, CA, USA), and the data were analyzed using BD CSampler Plus software.

### 4.6. Western Blot Assay

Protein samples were extracted from the cells with lysis buffer and then separated using 10% polyacrylamide gel before transfer to polyvinylidene fluoride (PVDF) membranes (Merck Millipore). The membranes were then blocked using 5% nonfat milk prepared in a TBST buffer for 1 h and subsequently incubated with primary antibodies for 24 h at 4 °C. Subsequently, the cells were incubated for 1 h with secondary (peroxidase-conjugated) antibodies at room temperature. Finally, the protein bands were assessed using an ImageQuant LAS 4000 Mini (GE Healthcare Life Sciences; Boston, MA, USA).

### 4.7. DAPI Staining

The cells were seeded onto 6-well plates (2 × 10^5^ cells/well) overnight and treated with various doses of SFB at 37 °C for 24 h. The cells were fixed with 4% (*v/v*) formaldehyde for 30 min at room temperature and permeabilized with 0.1% (*v/v*) Triton X-100. After washing twice in PBS, the cells were stained with DAPI (50 μg/mL) at room temperature for 15 min. The morphological changes were photographed by fluorescence microscope (Lecia, Bensheim, Germany).

### 4.8. Annexin V/PI Double Staining Assay 

As previously described [53], the cells were treated with different concentrations of SFB for 24 h. Then, the cells were harvested and suspended in PBS (2% BSA) and incubated with BD Annexin V and Dead Cell reagent for 15 min at room temperature in dark. The cells were analyzed by BD Accuri C6 Plus flow cytometry, and the data were analyzed using BD CSampler Plus software.

### 4.9. Mitochondrial Membrane Potential Measurement

As previously described [51], the cells were incubated with different concentrations of SFB for 24 h. The cells were then collected and stained with BD MitoPotential working solution at 37 °C for 20 min. After incubating the cells with 5 μL of 7-AAD for 5 min, BD Accuri C6 Plus flow cytometry was used to analyze the samples. The data were analyzed by BD CSampler Plus software.

### 4.10. Reactive Oxygen Species (ROS) Measurement

As previously described [54], the cells were incubated with different concentrations of SFB for 24 h. The cells were then collected and stained with DCFH-DA to measure ROS levels. After incubation, BD Accuri C6 Plus flow cytometry was used to analyze the samples. The data were analyzed by BD CSampler Plus software.

### 4.11. Caspase-3/7 Detection and Analysis

The analysis was performed as previously described [55]. The user guide of the Muse Caspase-3/7 kit describes the Caspase-3/7 detection method. After treatment with SFB, the cells were stained with Muse Caspase-3/7 reagent. The cells were visualized using a flow cytometer and analyzed using Muse Cell Soft.

### 4.12. Gene Transfection

The survivin overexpression plasmid and negative control plasmid were obtained from Sino Biological (Wayne, PA, USA). The cells were seeded onto a 6 cm dish and transfected with plasmid using TurboFect Transfection Reagent (Thermo Fisher Scientific; Waltham, MA, USA). The effects of survivin up-regulation were further measured using the Western blot assay.

### 4.13. Statistical Analysis

All statistical analyzes were performed with SigmaPlot v12.5 (Systat Software Inc.; Palo Alto, CA, USA). The data were collected from three independent experiments. One-way analysis of variance and Tukey’s multiple comparison test were performed to compare between the control and treatment cells. A *p* value of <0.05 was considered statistically significant.

## 5. Conclusions

The study describes the anticancer effect and mode of action of Glycyrrhiza-derived compound SFB in human OSCC. The study findings reveal that SFB reduces oral cancer cell viability by arresting cell cycle at the G2/M phase and inducing caspase-mediated apoptosis. Both intrinsic and extrinsic apoptotic pathways are affected by SFB in oral cancer cells. Moreover, SFB-mediated induction of ROS production plays a significant role in apoptosis induction. Regarding upstream signaling pathways, SFB has been found to reduce MAPK and Ras/Raf/MEK signaling to accomplish its pro-apoptotic effect. Notably, our human apoptosis array findings have identified survivin as the major apoptosis-related target of SFB. Taken together, the present study is the first of its kind to identify SFB as a potent anticancer agent.

## Figures and Tables

**Figure 1 ijms-24-04505-f001:**
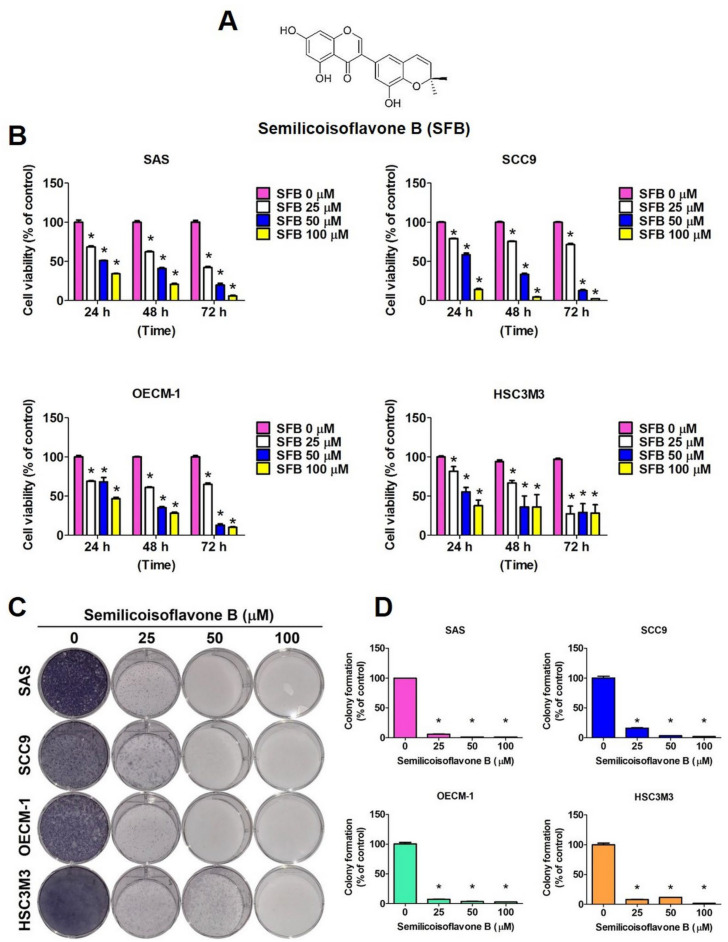
Toxicity of semilicoisoflavone B on oral cancer cells. (**A**) Chemical structure of semilicoisoflavone B (SFB). (**B**) Results of the MTT assay for oral cancer cells (SAS, SCC9, OECM-1 and HSC3M3) treated with indicated concentrations of semilicoisoflavone B for 24 h. (**C**,**D**) Results of the colony formation assay for oral cancer cells treated with indicated concentrations of semilicoisoflavone B after two weeks. Data are the mean ± standard deviation (SD) of three independent experiments. * *p* < 0.05, compared with the control group.

**Figure 2 ijms-24-04505-f002:**
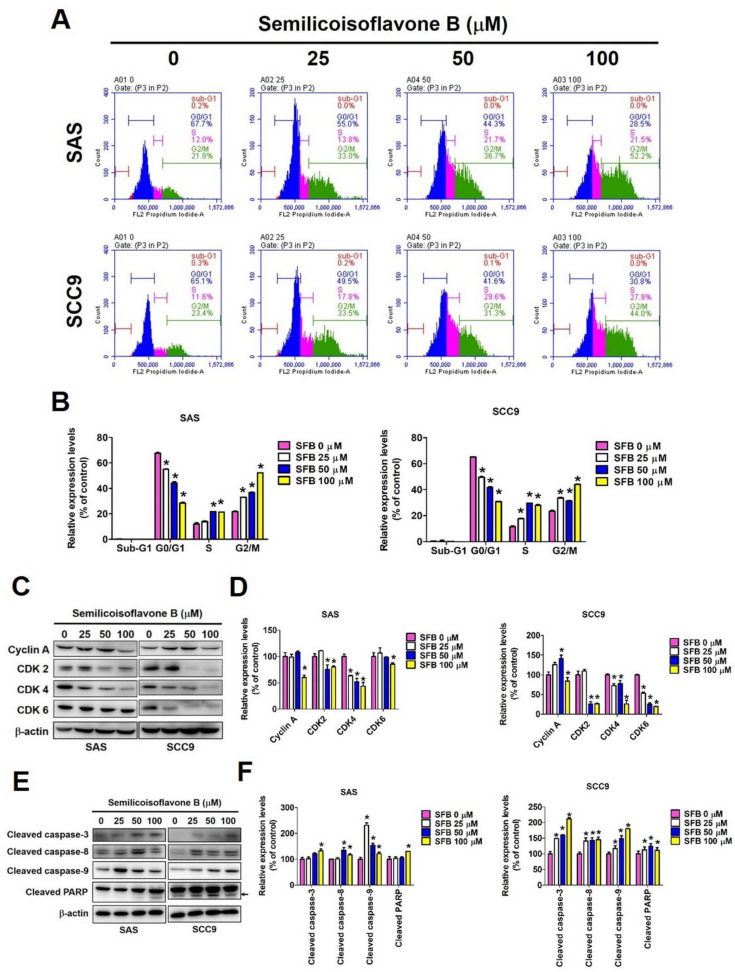
Semilicoisoflavone B induces cell cycle arrest and apoptosis in SAS and SCC9 cells. After treated SFB (0–100 μM) for 24 h. (**A**) Cells were used PI stained and flow cytometry performed to estimate cell cycle phase distribution. (**B**) Quantification of different cell cycle phase of sub-G1, G0/G1, S and G2/M. (**C**) Analyze the expression of cell cycle control proteins, including cyclin A, CDK 2, CDK 4, and CDK 6 by Western blotting. (**D**) Quantitative relative density of each protein level was normalized to β-actin. (**E**) Analyze the expression of cell cycle control proteins, including cleaved caspase-3, cleaved caspase-8, cleaved caspase-9, and cleaved PARP by Western blotting. (**F**) Quantitative relative density of each protein level was normalized to β-actin. Data are presented as mean ± SD (*n* = 3). * *p* < 0.05, compared with the control group.

**Figure 3 ijms-24-04505-f003:**
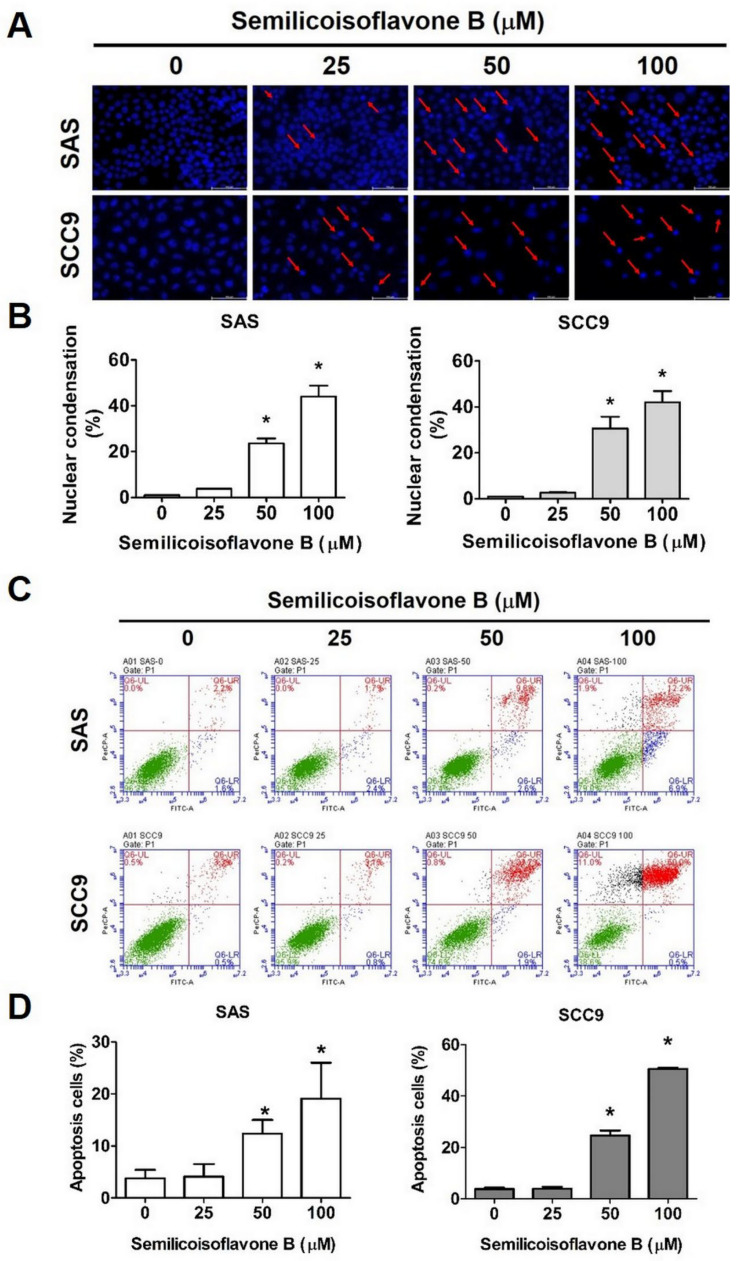
Semilicoisoflavone B induces apoptosis in SAS and SCC9 cells. After treated SFB (0–100 μM) for 24 h. (**A**) Used DAPI stain assay to determine DNA condensation with fluorescence microscopy. (**B**) Quantitative relative density of the percentage of nuclear condensation. (**C**) Cells were stained with Annexin V/PI and flow cytometry was used to reveal SFB-induced apoptosis. (**D**) Quantitative relative density of the percentage of apoptosis cells (including early and late state). Data are presented as mean ± SD (*n* = 3). * *p* < 0.05, compared with the control group.

**Figure 4 ijms-24-04505-f004:**
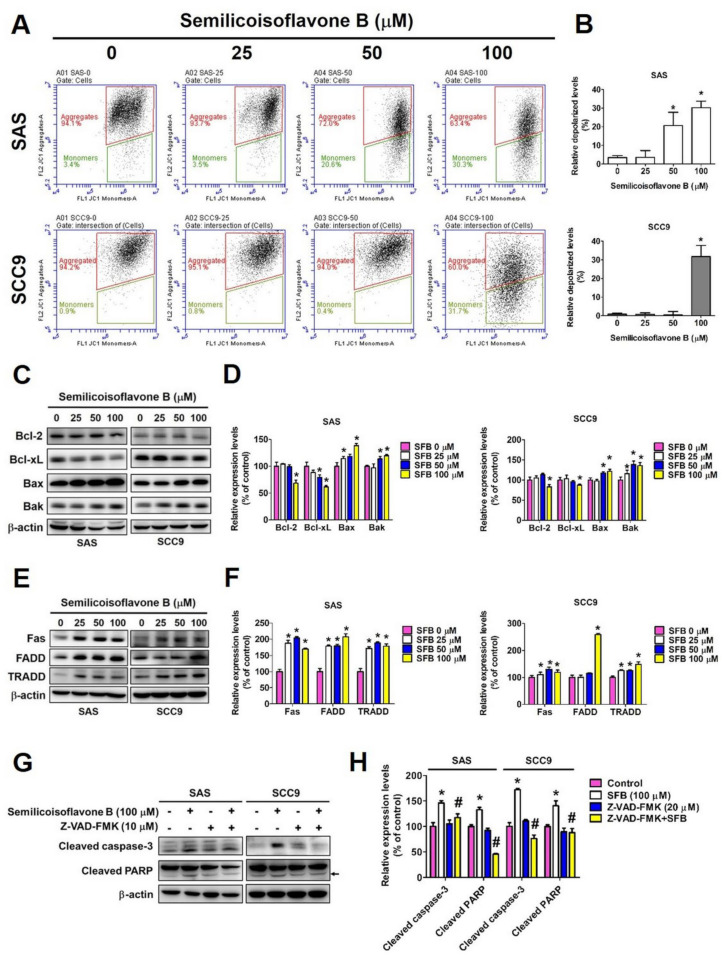
Intrinsic pathway and the extrinsic pathway were regulated by semilicoisoflavone B in oral cancer cell lines. After treated SFB (0–100 μM) for 24 h. (**A**) Mitochondrial membrane potential measurement assay was used by flow cytometry. (**B**) The data was analyzed by BD CSampler Plus software. (**C**) Analyze the expression of Bcl-2 family proteins, including Bcl-2, Bcl-xL, Bax and Bak by Western blotting. (**D**) Quantitative relative density of each protein level was normalized to β-actin. (**E**) Analyze the expression of intrinsic pathway control proteins, including Fas, FADD, and TRADD by Western blotting. (**F**) Quantitative relative density of each protein level was normalized to β-actin. (**G**) After combine treated SFB (100 μM) with/without Z-VAD-FMK (caspases inhibitor, 10 μM) for 24 h. Analyze the expression of apoptosis-related proteins, including cleaved caspase-3 and cleaved PARP by Western blotting. (**H**) Quantitative relative density of each protein level was normalized to β-actin. Data are presented as mean ± SD (*n* = 3). * *p* < 0.05, compared with the control group. # *p* < 0.05 compared with the SFB-only treatment group.

**Figure 5 ijms-24-04505-f005:**
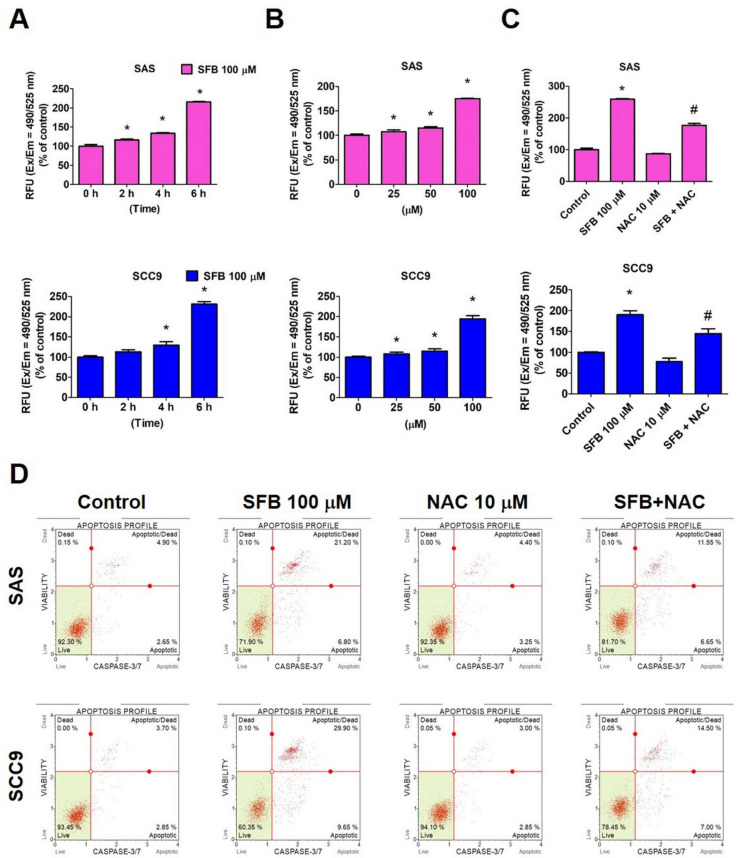
Semilicoisoflavone B induces apoptosis by ROS production in oral cancer cell lines. (**A**) After being treated with SFB (100 μM) for 0, 2, 4, and 6 h, ROS assay was used by flow cytometry. (**B**) After being treated with SFB (0–100 μM) for 6 h, ROS assay was used by flow cytometry. (**C**) After being treated with SFB (100 μM) with/without NAC (ROS inhibitor, 10 μM) for 6 h, the expression of ROS by flow cytometry was analyzed. (**D**) After being treated with SFB (100 μM) with/without NAC for 6 h, the expression of apoptosis cells by flow cytometry was analyzed. Data are presented as mean ± SD (*n* = 3). * *p* < 0.05, compared with the control group. # *p* < 0.05 compared with the SFB-only treatment group.

**Figure 6 ijms-24-04505-f006:**
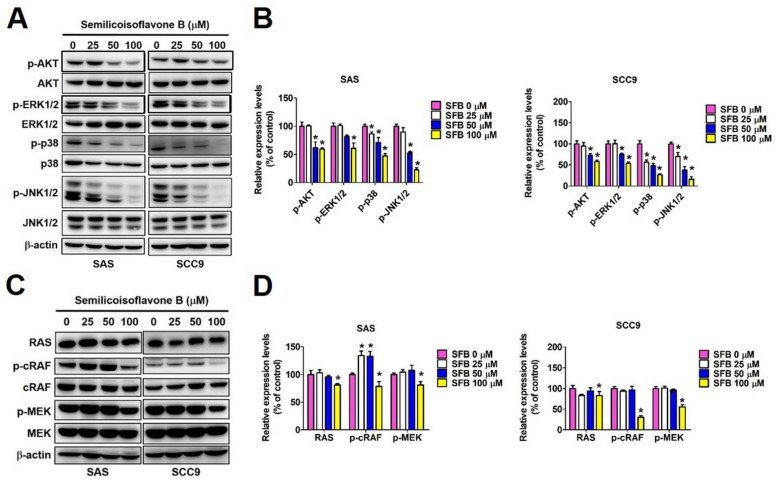
Semilicoisoflavone B induces apoptosis by affecting the Ras/Raf/MEK, PI3K/AKT and MAPK pathways in oral cancer cell lines. (**A**, **B**) Western blotting was used to measure the expression of AKT and MAPK pathway proteins and (**C**, **D**) the Ras/Raf/MEK pathway proteins. Quantitative relative density of each protein level was normalized to β−actin. Data are presented as mean ± SD (*n* = 3). * *p* < 0.05, compared with the control group.

**Figure 7 ijms-24-04505-f007:**
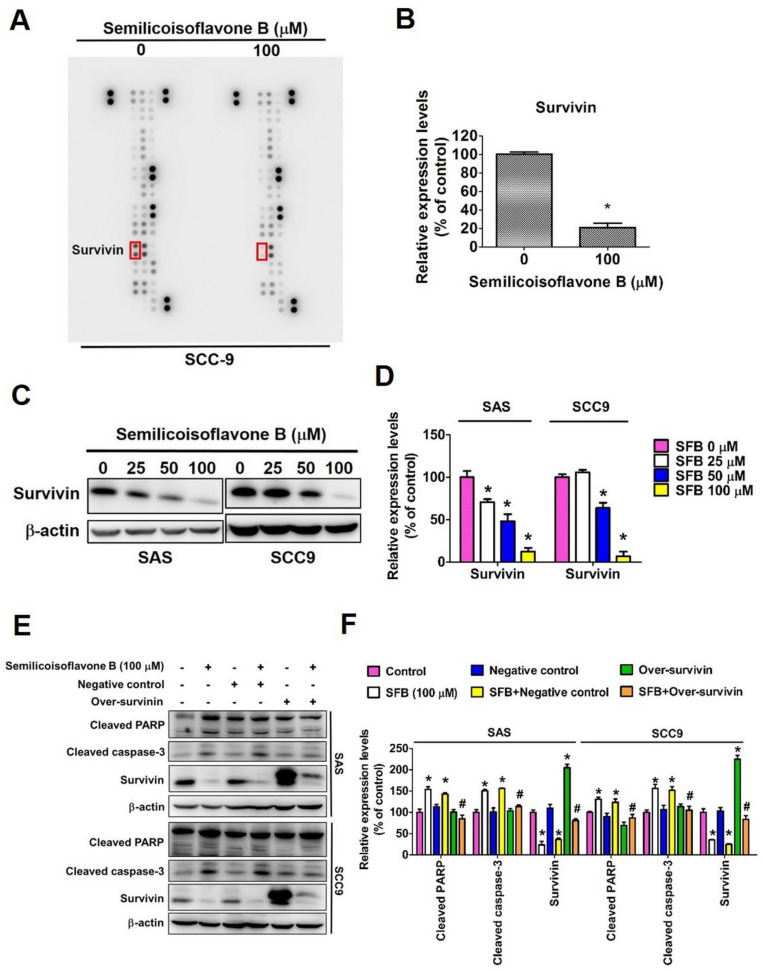
Survivin plays a crucial rule in semilicoisoflavone B-induced antiproliferation and PARP activation. (**A**) After treatment of 100 μM SFB for 24 h in SCC-9 cells, the human apoptosis array (ARY009, R&D systems) were employed and the decreased survivin protein was exposed to quantitative analysis. (**B**) Intensity qualifications of survivin in SFB-treated SCC-9 cells. (**C**) SAS and SCC9 cells were treated with the indicated doses of SFB (0, 25, 50, and 100 μM), and the survivin expression level was detected through Western blotting. The β-actin protein level was used to adjust quantitative results. (**D**) Graphs show the findings of the statistical analysis of survivin proteins. (**E**) SAS and SCC9 cells were cotreated with over-survivin plasmid and SFB (100 μM), and the cleaved caspase-3, cleaved PARP and survivin expression level was analyzed using Western blotting. The β-actin protein level was used to adjust quantitative results. (**F**) Graphs show the findings of the statistical analysis of cleaved caspase-3, cleaved PARP and survivin protein. Data are presented as the mean ± SD from three independent experiments, * *p* < 0.05 compared with the vehicle treatment group; # *p* < 0.05 compared with the SFB-only treatment group.

## Data Availability

This study did not report any data.

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
