# Peer review of "Semilicoisoflavone B Induces Apoptosis of Oral Cancer Cells by Inducing ROS Production and Downregulating MAPK and Ras/Raf/MEK Signaling"

_ijms, 2023, doi:10.3390/ijms24054505_

Round 1
Reviewer 1 Report
The authors demonstrated that semilicoiso-26 flavone B (SFB), which is a natural phenolic compound isolated from Glycyrrhiza species, had anti-cancer effects on oral squamous cell carcinoma (OSCC) cell lines. SFB decreased cell viability, induced apoptosis by both intrinsic and extrinsic pathways, increased ROS production. SFB also downregulated surviving, causing PARP and caspase-3 cleavage.
The manuscript is well organized, however, there are several concerns.
#1 Nuclear staining (DAPI) is dim. The condensation is not clear.
#2 The data of Figure 6 – pAKT and pERK was not suppressed by 25 and 50 microM SFB. In contrast, SFB decreased pp38 and pJNK in dose dependent manner. These sounds like different roles between these signaling molecules.
#3 Moreover, the sentence “these findings indicated that …(Page 10 line 218-219)” is exaggerated, since the authors did not show the relationship between signaling pathway and apoptosis at all.
#4 SFB downregulated survivin protein expression. How about the message of sruvivin (mRNA) by SFB treatment? What is the mechanism? At least, the authors should discuss this, since they citated several previous paper about survivin.
#5 There are many misspelled words and phrases. Please check the whole manuscript carefully and correct them.
For example,
Page 4 line 146 Fad à Fas
Page 4 line 149 caspase 3 à cleaved caspase 3
Page 5 line 161 PARP à cleaved PARP
Page 7 line 177 DR5, DcR3, … à FADD, and TRADD
Page 7 line 179 corrupted text à beta etc.
Reviewer 2 Report
Interesting and well written article
Author Response
Answers: Thanks for this valuable suggestion. It is our pleasure to receive such comments.Reviewer 3 Report
The present work addresses a topic of importance for clinical and basic research – the role of semilicoisoflavone B as an inducer of apoptosis in oral cancer cells.
Abstract is well structured and informative. Introduction is concise and correct. It summarizes the recent knowelege for OSCC and semilicoisoflavone B. The results are satisfactory. Their presentation is detailed and logically developed. Details are correct. Data shown are adequate to the aim and highly informative.
The few remarks are listed below:
1. The description of fig 1a should me mentioned also in the result section not only in the discussion.
2. How the authors could explain the reduction in the cell cycle rate at the G0/G1 phase and the increase at the S and G2/M phases?
3. The discussion could be expanded with interpretation of the results in aspect of ROS induced changes.
4. In order to see the mode of action of semilicoisoflavone B on normal cells, it would be interesting to perform some of the experiments related to cell cycle arrest, apoptosis and ROS on human oral epithelial cells.
In conclusion, the topic of the manuscript is very interesting and will stimulate the reader’s interest. Finally, I recommend the proposed manuscript to be accepted for publication at IJMS after answering the above mentioned corrections.
